# A Selective Multi-Branch Network for Edge-Oriented Object Localization and Classification

**Kai Su** *[ID], **Yoichi Tomioka** [ID], **Qiangfu Zhao** [ID] **and Yong Liu** [ID]

Graduate School of Computer Science and Engineering, The University of Aizu,
Aizu-Wakamatsu 965-8580, Japan; ytomioka@u-aizu.ac.jp (Y.T.); qf-zhao@u-aizu.ac.jp (Q.Z.);
yliu@u-aizu.ac.jp (Y.L.)
* Correspondence: d8232114@u-aizu.ac.jp

**Abstract:** This study introduces a novel selective multi-branch network architecture designed to speed up object localization and classification on low-performance edge devices. The concept builds upon the You Only Look at Interested Cells (YOLIC) method, which was proposed by us earlier. In this approach, we categorize cells of interest (CoIs) into distinct regions of interest (RoIs) based on their locations and urgency. We then employ some expert branch networks for detailed object detection in each of the RoIs. To steer these branches effectively, a selective attention unit is added into the detection process. This unit can locate RoIs that are likely to contain objects under concern and trigger corresponding expert branch networks. The inference can be more efficient because only part of the feature map is used to make decisions. Through extensive experiments on various datasets, the proposed network demonstrates its ability to reduce the inference time while still maintaining competitive performance levels compared to the current detection algorithms.

**Keywords:** object localization and classification; YOLIC; real-time detection; edge device

## 1. Introduction

With the rapid development of computer vision based on deep learning, developing efficient and accurate detection methods is crucial, especially in the context of Tiny AI [1]. The rise of smart devices such as micro-robots and smart IoT devices has highlighted the need for robust algorithms designed for such constrained settings. Beyond the fundamental challenges of energy efficiency and processing power, modern edge computing must also navigate critical issues like data privacy and the need for real-time processing [2,3]. Therefore, it is important to develop specialized detection methods for edge devices in this scenario.

The vision-based deep learning approaches for object detection can be roughly classified into two-stage approaches and one-stage ones. The two-stage methods (e.g., Faster RCNN [4], Cascade R-CNN [5], and D2Det [6]) are typically renowned for their notable detection accuracy, while the one-stage approaches (e.g., SSD [7] and YOLO series [8–13]) are generally characterized by their fast detection speed. These two classes of approaches differ in their detection processes and performance trade-offs.

In recent years, lightweight detection models designed specifically for edge devices have primarily been based on speed-oriented one-stage approaches. These models typically substitute the large backbone networks for object detection with lightweight networks or modules to facilitate rapid detection. For instance, PG-YOLO [14] replaces the convolutional modules in YOLOv5 [10] with low computational ghost modules [15], thereby reducing the computational complexity and resulting in faster inference. The study [16] replaces the original VGG network [17] in the original SSD framework with a lightweight network to reduce the model parameters and improve the detection efficiency in smart devices. Micro-YOLO [18] replaces several convolutional layers in the YOLOv3-tiny network [19] with lightweight depth-wise separable convolutional layers with squeeze and excitation

blocks [20], reducing the model size by 3.46 times and the multiply–accumulate operation (MAC) by 2.55 times compared to the original YOLOv3-tiny network. Liu et al. [21] introduced lightweight C3Ghost and GhostConv modules into the YOLOv5-S backbone network to compress the size of the model. As the demand for lightweight models continues to grow, a trend has emerged whereby new object detection projects include models tailored to lightweight devices. For example, YOLOv5-N [10], YOLOv6Lite [11], YOLOX-Nano [9], and YOLOv8-N [13] are designed to meet the efficiency requirements of low-resource environments. Despite these strategies significantly reducing the computational demands, lightweight detection models still struggle to satisfy the needs of low-performance edge devices while maintaining the detection accuracy.

In this study, we enhance the You Only Look at Interested Cells (YOLIC) method, which was proposed by us earlier [22], by introducing a selective multi-branch architecture. The main goal is to reduce the inference workload for low-performance devices. The basic idea is to group all cells of interest (CoIs) into distinct regions of interest (RoIs) based on the needs of the detection task (e.g., location and urgency) and focus on critical RoIs (e.g., RoIs close to the car and containing some risky objects) during inference to increase the detection speed. For this purpose, we add a selective attention unit (SAU) that can identify the critical RoIs at an earlier stage and enable the whole system to concentrate its detection efforts on the needed RoIs. By concentrating on the critical RoIs and disregarding unimportant parts of the image, the new approach is expected to improve the processing speed on edge devices. Additionally, the integration of selective feature map fusion within an in-network feature pyramid structure is a key innovation in our network. This allows the branches to concentrate specifically on the RoI feature map at different scales, thereby improving the network's accuracy and efficiency in real-time scenarios.

The contributions of this paper are summarized as follows.

- We introduce a novel selective multi-branch architecture that can reduce the mean inference workload on low-performance edge devices. This approach builds on the YOLIC approach to improve the network efficiency by only looking at the features of critical ROIs.
- We design a selective attention unit at an intermediate stage of the detection process to guide the activation of different branches. This unit is used to identify RoIs that may contain objects and focuses the detection effort on these areas in the subsequent stages of the process.
- We incorporate selective feature map fusion when features need to be fed into each branch network. This integration focuses on predefined cell locations within the feature map to identify RoIs. By aggregating the features of RoI areas across multiple scales and applying weighted integration, our selective multi-branch significantly reduces the computational burden.
- We provide experimental results to verify the effectiveness of the proposed selective multi-branch network using three datasets. The experimental results demonstrate that the proposed selective multi-branch network can effectively reduce the inference time, maintaining competitive performance compared with the previous method.

It is important to note that this paper is an expanded version of a work presented at a prior conference [23], but it includes significant additions and enhancements. The primary distinction lies in the introduction of a selective attention unit to our architecture, which efficiently identifies critical RoIs early in the detection process and guides the activation of different branches. Additionally, the robustness and efficacy of our proposed selective multi-branch network are further tested through experiments conducted on three different datasets.

The rest of this paper is organized as follows. Section 2 provides an overview of our previous research on YOLIC, attention mechanisms, multi-scale feature fusion techniques, and multi-branch structures. Section 3 details our selective multi-branch network architecture, including its block design, loss function, and implementation details. In Section 4, we present three detection tasks and detail the experimental results. Section 5 is dedicated to

discussing the outcomes of the comparative experiments. Finally, Section 6 concludes the paper and introduces our future work.

## 2. Related Work

### 2.1. You Only Look at Interested Cells

YOLIC [22] introduces an efficient detection method on edge devices in the Tiny AI domain. By integrating the strengths of semantic segmentation and object detection, YOLIC offers enhanced computational efficiency and precision. Central to its approach is the utilization of predefined CoIs for object localization and classification, moving away from the conventional focus on individual pixels. This technique encapsulates vital information while reducing the computational burden, and it also aids in approximating the rough shapes of objects. A key advantage of YOLIC is the elimination of the need for bounding box regression due to its reliance on predetermined cell configurations that inform about the potential location, size, and shape of objects. Another feature of YOLIC is its implementation of multi-label classification for each cell, effectively overcoming the challenges posed by single-label classification, particularly in cases with overlapping or adjacent objects. This method allows YOLIC to accurately identify multiple objects within a single cell. However, a limitation of this approach is that it processes extensive regions of the image that may not be pertinent, resulting in the inefficient use of computational resources during the detection of specific predefined areas.

### 2.2. Attention Mechanism

Attention mechanisms can effectively help a neural network to focus on the most relevant parts of the input to perform detection tasks. In the current object detection algorithms, an attention module is designed to focus on either spatial features, channel-related features, or a combination of both. This adaptability allows the network to dynamically give precedence to the most significant features. Squeeze-and-Excitation Attention [20] understands the channel interdependencies in a feature map and adjusts the map accordingly to enhance the important features. A convolutional block attention module [24] integrates both spatial and channel attention, refining the feature maps by focusing on important regions. A pyramid split attention block [25] efficiently captures multi-scale contextual information by splitting the feature maps and applying attention at various scales. The lightweight SGE module [26] processes each group's sub-features in parallel, employing global–local feature similarity as attention guidance. Recent studies [21,27,28] have also demonstrated that incorporating attention mechanisms into object detection algorithms can significantly enhance the model's detection performance across various applications. In contrast to these remarkable approaches, where attention modules are leveraged to find useful features, we design a selective attention unit to find important predefined regions and ignore the background areas in the subsequent process.

### 2.3. Multi-Scale Feature Fusion

Multi-scale feature fusion has become an indispensable component in leading detection algorithms. This technique integrates various features at different scales to create a more comprehensive representation of the features. Lin et al. [29] introduced a top-down architecture with lateral connections, establishing high-level semantic feature maps across various scales. The Path Aggregation Network [30] incorporated bottom-up path aggregation to effectively shorten the information path and enrich the feature pyramid with precise localization signals from lower-level features. NAS-FPN [31] employed a neural architecture search to discover an irregular feature fusion structure. BiFPN [32] introduced several optimizations for cross-scale connections, bringing a better balance between accuracy and efficiency. While newer feature fusion structures have led to improved detection accuracy, they also introduce considerable computational complexity due to the intricate feature integration. In many practical applications [33–35], the detection performance can be effectively improved by using fused multi-scale features. Considering the need for efficient detection methods

suitable for edge devices, this study proposes a selective multi-scale feature fusion structure based on the simplest FPN structure. In the proposed fusion structure, feature maps of each scale are selectively fused according to the CoI regions defined in the original input images.

### 2.4. Multi-Branch Structure

The multi-branch structure is prominently featured in advanced detection algorithms like the YOLO series [11,13]. Typically, these algorithms employ a backbone network for common feature extraction. Each branch within the structure then processes the different scales of feature maps outputted by the backbone's feature pyramid. This approach allows accurate predictions for objects of different sizes, leveraging the distinct capabilities of each branch to handle specific feature scales. YOLOX [9] utilizes this idea to predict bounding boxes and class probabilities separately. This strategy enhances the overall detection performance. In various real-world scenarios, networks with multi-branch architectures have found extensive application, such as pedestrian detection [36], defect detection [37], and medical-image-assisted diagnosis [38]. Contrary to the multi-branch structures in current detection algorithms, which construct multi-branch structures by adding extra layers outside of the backbone network, such an approach is not feasible for edge-device-oriented lightweight algorithms. To address this, we propose a solution: decomposing a single backbone network into a multi-branch structure without adding extra components. This design maintains the integrity and efficiency of a single backbone while taking advantage of a multi-branch architecture.

### 3. Selective Multi-Branch Network

The proposed selective multi-branch detection network comprises two essential components: a common feature extraction block and multiple sub-branch blocks. In Figure 1, we present the detailed architecture of our network. The proposed network is crafted by modifying a single lightweight MobileNet-v2 network [39]. By segmenting the original MobileNet-v2 model, we form the central feature extraction block and multiple sub-branch blocks, each derived from the MobileNet-v2 architecture, without adding extra computational layers. Notably, a combination of one feature extraction block and one sub-branch block constitutes the original MobileNet-v2 structure. This strategy ensures that our selective multi-branch network not only harnesses the efficiency and lightweight characteristics of MobileNet-v2 but also maintains a compact and effective framework suitable for edge computing. The detailed division and definition of the network structure are outlined in Table 1. In the following paragraph, we will delve into the specifics of each part of our network.

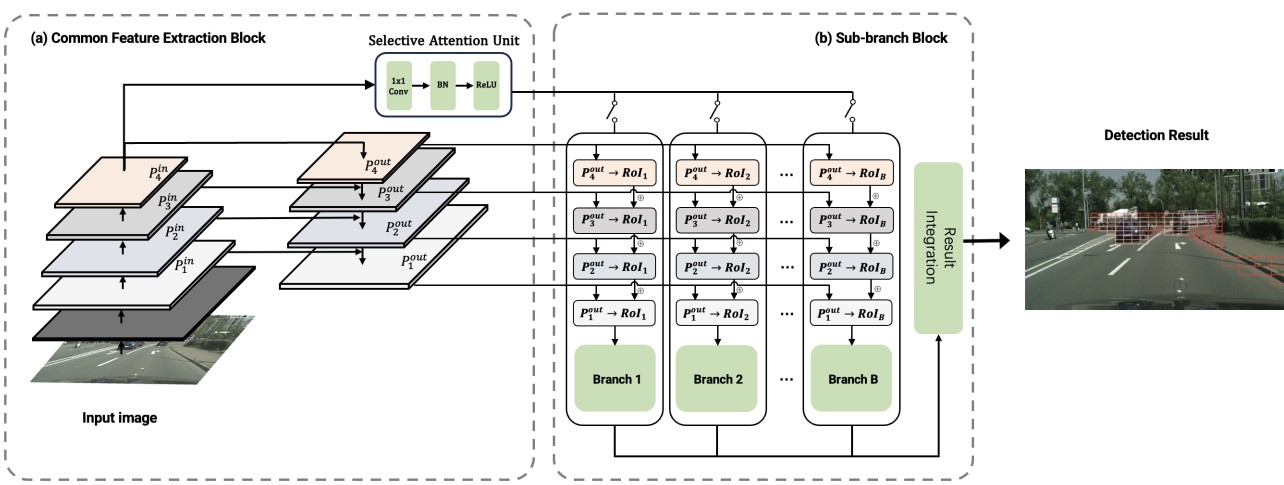

**Figure 1.** The architectural overview of the proposed selective multi-branch network. Panel (**a**) shows the common feature extraction block, where input images are processed through multiple layers to

produce feature maps, which are then refined through the selective attention unit. Panel (**b**) depicts the sub-branch block, where each feature map is cropped based on RoIs and processed through individual branches (Branch 1 to Branch B). The $\oplus$ symbol depicted in the diagram represents a weighted add operation, meaning that the RoI features from each pyramid layer are combined together with different weights.

**Table 1.** The detailed specifications of the selective multi-branch network architecture. This table enumerates the sequence of layers within the common feature extraction block and N sub-branch blocks, outlining the layer types, output sizes, filter dimensions, strides, expansion factor (t), channel size (c), number of repeated layers (n), and final predictions ($C_i$).

| Block | Layer | Output Size | Stride | t | c | n |
|---|---|---|---|---|---|---|
| | Input Layer | $224 \times 224 \times 3$ | - | - | 3 | 1 |
| Common | Standard Convolution | $112 \times 112 \times 32$ | 2 | - | 32 | 1 |
| Feature | Inverted Residual Block | $112 \times 112 \times 16$ | 1 | 1 | 16 | 1 |
| Extraction | Inverted Residual Block | $56 \times 56 \times 24$ | 2 | 6 | 24 | 2 |
| Block | Inverted Residual Block | $28 \times 28 \times 32$ | 2 | 6 | 32 | 3 |
| | Inverted Residual Block | $14 \times 14 \times 64$ | 2 | 6 | 64 | 4 |
| | Inverted Residual Block | $14 \times 14 \times 96$ | 1 | 6 | 96 | 3 |
| | Inverted Residual Block | $7 \times 7 \times 160$ | 2 | 6 | 160 | 3 |
| B | Inverted Residual Block | $7 \times 7 \times 320$ | 1 | 6 | 320 | 1 |
| Sub-Branch | Pointwise Convolution | $7 \times 7 \times 1280$ | 1 | - | 1280 | 1 |
| block | Global Average Pooling | $1 \times 1 \times 1280$ | - | - | 1280 | - |
| | Dropout Layer | $1 \times 1 \times 1280$ | - | - | 1280 | - |
| | Fully Connected Layer | $C_i$ | - | - | $C_i$ | - |

*3.1. Common Feature Extraction Block*

The common feature extraction block in our branch Network is derived from the initial five stages of the MobileNet-v2 architecture. Its primary function is to perform the preliminary feature extraction from the entire input image. This process involves capturing essential image features necessary for effective subsequent detection. To enrich the features provided to the subsequent networks, we have implemented a pyramid structure following the methodology described in [29]. This structure capitalizes on the outputs from each stage of the common feature extraction block to utilize both the semantic information of high-level features and the detailed information of shallow features. The result of this in-network pyramid structure is the production of four different sizes of feature maps, denoted as $P_1^{out}$ to $P_4^{out}$, which are derived from the corresponding outputs ($P_1^{in}$ to $P_4^{in}$) of each stage in the common feature extraction block. The process of generating these outputs is as follows:

$$
\begin{aligned}
P_4^{out} &= Conv(P_4^{in}) \\
P_3^{out} &= Conv(P_3^{in} + Resize(P_4^{out})) \\
P_2^{out} &= Conv(P_2^{in} + Resize(P_3^{out})) \\
P_1^{out} &= Conv(P_1^{in} + Resize(P_2^{out}))
\end{aligned}
\tag{1}
$$

where *Conv* is a $1 \times 1$ convolutional layer used to align the feature channels, and *Resize* refers to the up-sampling operation.

In the final output layer of the common feature extraction block, we incorporate a selective attention unit. This unit focuses on RoIs where multiple adjacent CoIs are predefined in the network. This unit's purpose is to identify potential locations of objects of interest within the input image. In an effort to maintain the compactness of the common feature extraction block, and given that this unit only needs to broadly identify regions with potential objects, we have chosen to utilize only the top-level feature P4 as the input for this unit. The processing flow of this unit can be observed in Figure 1a. Structurally,

the selective attention unit is kept streamlined. It consists of a convolutional layer, batch normalization, a ReLU layer, and a fully connected layer. This unit outputs region attention scores for each RoI, reflecting the likelihood of each region containing an object of interest. The number of these output values is in alignment with the number of sub-branch blocks in our network. When an RoI's attention score exceeds a certain threshold, the corresponding sub-branch block is activated for precise object localization and classification on its CoIs. Conversely, if the attention score is below the threshold, the unit can directly output the detection results, thereby reducing the time and computational resources spent in scanning irrelevant areas of the network.

*3.2. Sub-Branch Block*

The sub-branch block is primarily tasked with the precise classification of the CoIs in activated RoIs. Each sub-branch block within the network is constructed from stages 6 to 11 of the MobileNet-v2 architecture. To accommodate diverse detection tasks and regional requirements, we replicate this block structure multiple times across the network. This replication allows the network to adeptly handle varying detection demands in different areas of the input image. It is worth noting that in our selective multi-branch network, the number of branches is set to be the same as the number of RoIs in the detection task. This design ensures that each RoI has a dedicated branch for processing.

A key challenge in designing the selective multi-branch network is to ensure the computational efficiency, crucial for deployment on low-performance edge devices. While each branch receives different scales of features from the common feature pyramid as inputs, processing all features in every branch would significantly increase the overall computational load. To address this, we employ an approach whereby each branch crops and processes only the features within its assigned RoI from the feature maps. This selective cropping markedly reduces the computational burden for each branch. In the process of fusing selective multi-scale features from the feature pyramid, we recognize that each branch's task or object size varies, causing different scales of features to contribute unequally to different branches. To address this issue, we introduce a feature weighting mechanism for each scale of selective features. This approach enables each sub-branch to learn the importance of each input feature scale.

The final feature representation for each sub-branch is computed by the weighted concatenation of selectively cropped features from different levels of the feature pyramid, which can be formulated as

$$Input_i = \sum_{f=1}^{4} W_i^{P_f} \times Crop(P_f^{out}, RoI_i) \tag{2}$$

where $Input_i$ represents the final feature input for a sub-branch assigned to $RoI_i$, $W_i^{P_f}$ represents the weight specific to $RoI_i$ for the $f_{th}$ layer of the feature pyramid, and $Crop(P_f^{out}, RoI_i)$ is the cropped output from the $f_{th}$ layer of the feature pyramid of $RoI_i$. The cropping operation is performed by calculating the coordinates of $RoI_i$ (i.e., $[x_i, y_i, w_i, h_i]$, where $(x_i, y_i)$ is the coordinate of the top-left corner, $w_i$ is the width, and $h_i$ is the height) in the original input image, scaling their sizes to match the spatial dimensions of the feature map $P_f^{out}$, and then extracting a rectangular region from the feature map that corresponds to the spatial extent of $RoI_i$. This process is applied to each layer of the feature pyramid, yielding a set of feature maps that are spatially aligned with $RoI_i$. These cropped feature maps are then combined using the weights $W_i^{P_f}$ learned by the model to form the final feature input $Input_i$ for the sub-branch assigned to $RoI_i$. Importantly, these weights are not static but are network parameters that can be learned and optimized during the training of each sub-branch. This learning process allows the network to dynamically adjust the contribution of each feature layer. The process of generating the input for sub-branch *i* is illustrated in Figure 2.

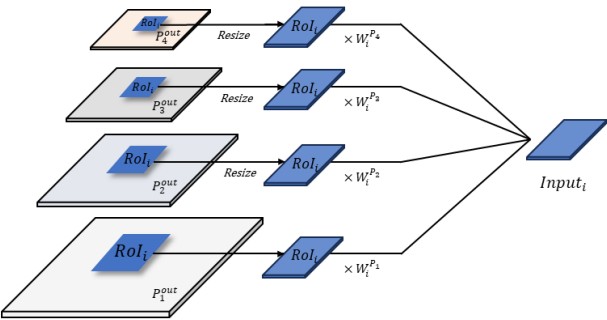

**Figure 2.** Illustration of the process of generating the input features for each sub-branch. The RoI coordinates are used to crop the corresponding regions from each layer of the feature pyramid. These cropped feature maps, after resizing, are combined using weights learned by the model specific to each RoI to form the final feature input for the sub-branch assigned to that RoI.

A single sub-branch is dedicated to analyzing a specific region designated by the selective attention unit. Assuming that a given region $i$ contains $N_i$ CoIs and $M_i$ objects of interest, a sub-branch parameterized by $\theta$ focuses its processing on this region. The output of this sub-branch $C_i$ is computed as follows:

$$C_i = Branch(Input_i; \theta_i) \tag{3}$$

where the length of $C_i$ is given by:

$$L_{C_i} = N_i \times (M_i + 1) \tag{4}$$

where $M_i + 1$ stands for the number of objects of interest within each CoI plus one background class. This output displays the detailed classification results obtained by a sub-branch for region $i$, which are generated only when this sub-branch is activated by the selective attention unit. The final detection outcome for the entire input image is the integration of the outputs from all sub-branches that have been activated by the attention unit. The decision regarding which sub-branch to activate is determined solely by the output of the attention unit. Our attention unit is responsible for focusing on the important regions and guiding the sub-branches to process these regions accordingly. Regions not highlighted by the attention unit bypass detailed examination.

### 3.3. Training Phase

The proposed selective multi-branch network is designed as an end-to-end system. Upon training with image inputs, all blocks can be learned from the teacher signal. There is no need for manual intervention in any of the modules throughout the process. The loss function used in our network is critical in guiding all output modules (including the selective attention unit and all sub-branch blocks). The overall loss function is a combination of the losses from all branches and the attention unit,

$$Loss_{all} = mean(L_{branches}) + \lambda L_{unit} \tag{5}$$

where the branch loss $L_{branches}$ is defined as

$$L_{branches} = \sum_i^B \sum_x^{N_i} \sum_y^{M_i+1} BCE(q_{xy}^i, p_{xy}^i) \tag{6}$$

and the attention unit loss $L_{unit}$ is given by

$$L_{unit} = \frac{1}{B} \sum_i^B BCE(t_i, k_i) \tag{7}$$

In these equations, $L_{branch}$ accounts for the performance of each sub-branch in accurately classifying the objects and cells of interest, while $L_{unit}$ evaluates the effectiveness of the selective attention unit in correctly identifying regions of interest. $mean(L_{branches})$ indicates the average loss across all the branches. The parameter $\lambda$ is a weighting factor that balances the contribution of the attention unit's loss to the overall loss. In this study, we have set this parameter to 0.02. This means that the weight of the loss from the attention unit in the overall loss is relatively small, which allows our network to primarily focus on the classification performance of each sub-branch. $B$ represents the total number of branches within the network, $N_i$ is the number of CoIs in the i-th region, while $M_i + 1$ represents the number of objects of interest plus a background class in the i-th region. $BCE$ stands for the binary cross-entropy loss function. $q^i_{xy}$ is the predicted probability for a particular cell $x$ and object $y$ in the i-th region and $p^i_{xy}$ represents the ground truth for the corresponding cell and object in the i-th region. Both $t$ and $k$ are vectors composed of $B$ elements. The element $t_i$ in vector $t$ represents the target value for the i-th region as identified by the attention unit. It serves as an indicator, specifying whether the i-th branch is activated for processing or not. On the other hand, the element $k_i$ in vector $k$ represents the predicted value for the same i-th region.

Algorithm 1 outlines the detailed training procedure for the selective multi-branch network. Initially, a MobileNet-v2 model is pre-trained on our target training dataset $\mathcal{D}$, ensuring that its components start with a robust feature extraction capability. Line 2 involves constructing the selective multi-branch network *Net* by integrating these pre-trained components into its feature extraction block and B sub-branch blocks. The next step is to set the network to training mode so that the model can output the results of all branches. In lines 4 and 5, we focus on determining the ground truths for both the attention unit and the branches. Ground truth $\mathcal{T}_u$ for the attention unit and $\mathcal{T}_b$ for the branches, ranging from 1 to $i$, are converted from the teacher signal $\mathcal{T}$. This extraction is essential in training the attention unit block and each sub-branch block on the specified tasks. The main loop of the algorithm iterates over each datum in the training dataset $\mathcal{D}$. In this loop, the data are passed through the selective multi-branch network, generating outputs from both the attention unit and all branches. The loss for the attention unit $Loss_{unit}$ and the loss for the branches $Loss_{branches}$ are calculated based on their respective ground truths. The total loss $Loss_{all}$ is then computed as the weighted sum of these two losses, with $\lambda$ acting as the weight for the attention unit's loss. Finally, this total loss is used for backpropagation through the network to update its parameters.

---

**Algorithm 1:** Training procedure for the selective multi-branch network

---

    **Input:** Training dataset $\mathcal{D}$ and teacher signal $\mathcal{T}$
    **Output:** Trained selective multi-branch network

1  **Initialize:** Pre-train MobileNet-v2 components on dataset $\mathcal{D}$
2  **Construct:** Selective multi-branch network *Net* by integrating pre-trained
    MobileNet-v2 components into its feature extraction and $B$ sub-branch blocks.
3  **Set Mode:** Configure *Net* to training mode.
4  **Determine Ground Truth:** Extract ground truth $\mathcal{T}_u$ for the attention unit from $\mathcal{T}$.
5  **Determine Ground Truth:** Extract ground truth $\mathcal{T}_b$ for branches 1 to i from $\mathcal{T}$.
6  **foreach** *data in $\mathcal{D}$* **do**
7      $Unit, Branches = Net(data)$
8      $Loss_{unit} = L_{unit}(\mathcal{T}_u^{data}, Unit)$
9      $Loss_{branches} = L_{branches}(\mathcal{T}_b^{data}, Branches)$
10     $Loss_{all} = mean(L_{branches}) + \lambda L_{unit}$
11     $Backpropagation(Loss_{all})$
12     $UpdateParameter(m)$
13 **end**

---

*3.4. Inference Phase*

During the inference phase, the selective attention unit plays a pivotal role in dynamically adjusting the activation of branch networks. This mechanism is key in enhancing the computational efficiency by avoiding the need to engage the entire network for every detection task. A important factor influencing the model's complexity and computational demand is the threshold value within the attention unit, which has been set at 0.5 for this study. This threshold is paramount in determining whether subsequent branches are activated, thereby maintaining a balance between the detection accuracy and inference speed. In future work, we plan to explore methods for the selection of the optimal threshold value based on the input data and the desired balance between accuracy and efficiency.

The inference process begins with the input image undergoing initial processing by a common feature extraction block. Following this, the network computes an attention score for each RoI. These scores are critical in determining which regions require deeper analysis. If an attention score for a given RoI exceeds the threshold, this RoI is escalated for further examination by the corresponding branch, as dictated by Equation (3). It is important to note that, initially, the output vector ($C$) for all RoI areas is initialized to signify the absence of any detected objects. Upon surpassing the threshold, the detection outcome for region $i$, denoted as $C_i$, is obtained, where $C_i \in \mathbb{R}^{N_i \times (M_i+1)}$, indicating that each branch's output is tailored to the specific RoIs that it processes. For regions that do not trigger branch activation based on their attention scores, all contained CoIs are classified as normal, suggesting that no obstacles are present within these sectors.

The end result of the branch detection process is to aggregate the results of all RoIs to form a comprehensive detection result. The final output of the selective multi-branch network can be represented as

$$\text{Final output} = \bigoplus_{i=1}^{B} C_i \tag{8}$$

where $C_i$ is defined by Equation (3), $\oplus$ denotes the concatenation operation, and the final output is a vector in $\mathbb{R}^{\sum_{i=1}^{B} N_i \times (M_i+1)}$.

Furthermore, this design concept significantly benefits edge devices, allowing them to reduce their computing power consumption and thus extend their battery life, especially in situations wherein the detection needs are reduced, such as during nighttime standby. Under ideal conditions, i.e., there are no obstacles within all RoIs, our model can achieve the fastest inference results without activating any branches. In contrast, in complex scenes where accurate object recognition is required, the model may need to activate all branches to ensure accuracy. This dynamic approach not only improves the efficiency but also ensures that our model remains robust and versatile across different detection scenarios.

## 4. Experiments and Evaluation

In this section, we describe experiments on three different datasets to demonstrate the efficiency of the proposed selective multi-branch network. The core of our experimental analysis lies in road obstacle detection. Recognizing the vital need for efficient and accurate obstacle detection in low-powered vehicles, our experiments aim to address this gap. All experiments were conducted using the PyTorch framework with an NVIDIA RTX 4090 GPU. Figure 3 provides an overview of our experimental workflow. The experimental dataset was divided into training, validation, and testing sets. We utilized the NVIDIA GPU to accelerate the training and evaluation of the proposed network implemented in PyTorch.

We employed the Adam optimizer for network training, starting with an initial learning rate of 0.001. A MultiStepLR scheduler was utilized to adjust the learning rate with milestones set at the 100-th and 125-th epochs. Our model was trained with input image resolutions of 224 × 224 over 300 epochs using a batch size of 32. To enhance the robustness of the model, we incorporated data augmentation techniques including random horizontal flipping and color jittering. In the evaluation, we compare our model with the state-of-the-

art detection algorithms (e.g., YOLOv5 [10], YOLOv6 [11], YOLOv8 [13]) using common metrics such as precision, recall, and the $F_1$-score for each object category. In the speed comparison experiments, we aimed to implement our models on resource-constrained and affordable edge devices to make the whole system more accessible to ordinary users. Therefore, we selected the Raspberry Pi 4B as our test platform to conduct speed evaluations at the edge. The inference framework used for the speed benchmark was ncnn, which is a dedicated neural network inference framework for mobile and embedded devices.

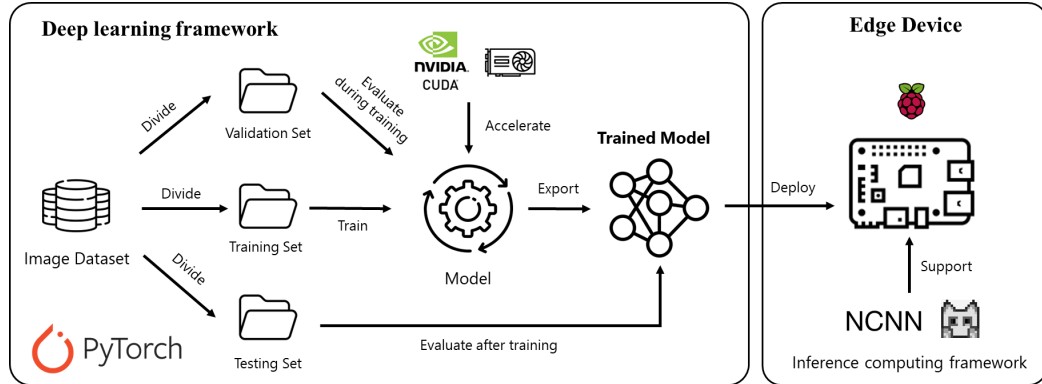

**Figure 3.** Overview of the experimental framework and workflow. The image dataset is divided into training, validation, and testing sets using PyTorch's data handling utilities. The NVIDIA GPU is used to accelerate the training and evaluation of the selective multi-branch network implemented in PyTorch. The trained model is then exported and deployed on edge devices, such as the Raspberry Pi 4B, for inference speed testing. The edge devices utilize the NCNN inference computing framework to support the efficient execution of the model in resource-constrained environments.

*4.1. Outdoor Hazard Detection*

The outdoor hazard detection experiment is primarily designed for low-cost electric scooters to develop an efficient and real-time detection system to ensure safety risk alerts. Following our previous work [22], we utilize the same cell configuration to design a detection range of 0–6 m on the road. This range is crucial in detecting obstacles in a scooter's immediate path. Within this area, we set up 96 CoIs based on the travel distance to accurately locate objects. Additionally, an area on the upper part of the image was designated for traffic sign detection. This region utilized 8 CoIs to roughly pinpoint the locations of traffic signs. To validate our proposed selective multi-branch network, we divided these 104 CoIs into four RoIs and constructed a selective multi-branch network with four corresponding branches. The division of the RoIs also was based on the distance: the first 32 CoIs within one meter were designated as RoI 1, handled by the first branch for training and detailed detection. The next 32 CoIs at 1–2 m formed RoI 2 for the second branch, while the 32 CoIs at 4–6 m constituted RoI 3 for the third branch. The traffic sign detection area's 8 CoIs were categorized as RoI 4, assigned to the fourth branch. The attention unit preliminarily assessed these four RoI areas. Figure 4 illustrates the division of the 104 RoIs and the four distinct areas.

Our dataset encompassed 20,380 outdoor images, each image annotated with common outdoor objects. We defined 11 types of objects for detection: Bump, Column, Dent, Fence, People, Vehicle, Wall, Weed, Zebra Crossing, Traffic Cone, and Traffic Sign. Following the partitioning method used in [22], the dataset was split into 70% for training, 10% for validation, and the remaining 20% for testing.

Table 2 presents a comprehensive comparison between our proposed selective multi-branch network models and the existing YOLO algorithm. This comparison is crucial to demonstrate the advancements and improvements that our model offers over the current state-of-the-art methods. Additionally, to evaluate the practical applicability of our network in real-world, resource-constrained environments, we conducted a speed comparison on a Raspberry Pi 4B, as detailed in Table 3.

**Table 2.** Detailed comparison of outdoor hazard detection performance for various objects of interest. Precision and recall scores of each model are presented. Detection performance metrics for YOLO are sourced from [22].

| Method | Input Size | Bump | | | Column | | | Dent | | | Fence | | | People | | | Vehicle | | |
|---|---|---|---|---|---|---|---|---|---|---|---|---|---|---|---|---|---|---|---|
| | | Precision | Recall | $F_1$-Score | Precision | Recall | $F_1$-Score | Precision | Recall | $F_1$-Score | Precision | Recall | $F_1$-Score | Precision | Recall | $F_1$-Score | Precision | Recall | $F_1$-Score |
| YOLOv5-N | 640 × 640 | 0.454 | 0.560 | 0.501 | 0.241 | 0.267 | 0.253 | 0.813 | 0.158 | 0.265 | 0.094 | 0.304 | 0.143 | 0.807 | 0.590 | 0.682 | 0.539 | 0.626 | 0.579 |
| YOLOv5-S | 640 × 640 | 0.360 | 0.678 | 0.470 | 0.147 | 0.564 | 0.233 | 0.423 | 0.041 | 0.075 | 0.117 | 0.132 | 0.124 | 0.822 | 0.414 | 0.551 | 0.557 | 0.695 | 0.618 |
| YOLOv6-N | 640 × 640 | 0.599 | 0.460 | 0.520 | 0.088 | 0.420 | 0.145 | 0.700 | 0.230 | 0.346 | 0.122 | 0.415 | 0.189 | 0.428 | 0.581 | 0.493 | 0.383 | 0.384 | 0.383 |
| YOLOv8-N | 640 × 640 | 0.867 | 0.793 | 0.828 | 0.662 | 0.718 | 0.689 | 0.872 | 0.681 | 0.765 | 0.831 | 0.371 | 0.513 | 0.892 | 0.850 | 0.870 | 0.872 | 0.879 | 0.875 |
| YOLOv8-S | 640 × 640 | 0.364 | 0.864 | 0.512 | 0.299 | 0.473 | 0.366 | 0.760 | 0.408 | 0.531 | 0.060 | 0.099 | 0.075 | 0.857 | 0.588 | 0.697 | 0.775 | 0.661 | 0.713 |
| YOLIC-M2 | 224 × 224 | 0.915 | 0.890 | 0.902 | 0.800 | 0.769 | 0.784 | 0.894 | 0.850 | 0.871 | 0.890 | 0.824 | 0.856 | 0.846 | 0.815 | 0.830 | 0.898 | 0.901 | 0.899 |
| Multi-branch | 224 × 224 | 0.908 | 0.882 | 0.895 | 0.807 | 0.750 | 0.777 | 0.872 | 0.830 | 0.850 | 0.892 | 0.769 | 0.826 | 0.844 | 0.823 | 0.834 | 0.896 | 0.897 | 0.897 |
| Selective multi-branch | 224 × 224 | 0.910 | 0.868 | 0.889 | 0.811 | 0.738 | 0.773 | 0.876 | 0.819 | 0.846 | 0.892 | 0.754 | 0.817 | 0.844 | 0.818 | 0.831 | 0.898 | 0.887 | 0.892 |

| Method | Input Size | Wall | | | Weed | | | Zebra Crossing | | | Traffic Cone | | | Traffic Sign | | | All | | |
|---|---|---|---|---|---|---|---|---|---|---|---|---|---|---|---|---|---|---|---|
| | | Precision | Recall | $F_1$-Score | Precision | Recall | $F_1$-Score | Precision | Recall | $F_1$-Score | Precision | Recall | $F_1$-Score | Precision | Recall | $F_1$-Score | Precision | Recall | $F_1$-Score |
| YOLOv5-N | 640 × 640 | 0.365 | 0.406 | 0.384 | 0.637 | 0.786 | 0.704 | 0.842 | 0.147 | 0.250 | 0.095 | 0.307 | 0.145 | 0.000 | 0.000 | 0.000 | 0.444 | 0.377 | 0.408 |
| YOLOv5-S | 640 × 640 | 0.213 | 0.754 | 0.332 | 0.563 | 0.923 | 0.699 | 0.548 | 0.537 | 0.542 | 0.015 | 0.015 | 0.015 | 0.000 | 0.000 | 0.000 | 0.342 | 0.432 | 0.382 |
| YOLOv6-N | 640 × 640 | 0.325 | 0.440 | 0.374 | 0.408 | 0.431 | 0.419 | 0.785 | 0.573 | 0.662 | 0.138 | 0.726 | 0.232 | 0.271 | 0.019 | 0.035 | 0.386 | 0.425 | 0.405 |
| YOLOv8-N | 640 × 640 | 0.839 | 0.833 | 0.836 | 0.841 | 0.898 | 0.869 | 0.882 | 0.808 | 0.843 | 0.780 | 0.630 | 0.697 | 1.000 | 0.018 | 0.036 | 0.849 | 0.680 | 0.755 |
| YOLOv8-S | 640 × 640 | 0.660 | 0.716 | 0.687 | 0.636 | 0.923 | 0.753 | 0.782 | 0.581 | 0.667 | 0.228 | 0.155 | 0.185 | 0.000 | 0.000 | 0.000 | 0.493 | 0.497 | 0.495 |
| YOLIC-M2 | 224 × 224 | 0.937 | 0.922 | 0.929 | 0.931 | 0.918 | 0.924 | 0.973 | 0.958 | 0.965 | 0.851 | 0.767 | 0.807 | 0.845 | 0.712 | 0.773 | 0.889 | 0.848 | 0.868 |
| Multi-branch | 224 × 224 | 0.934 | 0.905 | 0.920 | 0.923 | 0.917 | 0.920 | 0.971 | 0.954 | 0.963 | 0.828 | 0.760 | 0.793 | 0.918 | 0.777 | 0.841 | 0.890 | 0.842 | 0.866 |
| Selective multi-branch | 224 × 224 | 0.936 | 0.897 | 0.916 | 0.927 | 0.907 | 0.917 | 0.975 | 0.941 | 0.957 | 0.832 | 0.748 | 0.788 | 0.931 | 0.628 | 0.750 | 0.894 | 0.819 | 0.855 |

**Table 3.** Evaluation of inference time (ms) on Raspberry Pi 4B with outdoor hazard dataset, including selective multi-branch model, with performance range indicated from minimum (no branch usage) to maximum load (full branch usage).

| Without Branch | One Branch | Two Branches | Three Branches | All Branches |
| --- | --- | --- | --- | --- |
| 54.98 | 59.09 | 66.22 | 71.84 | 74.82 |

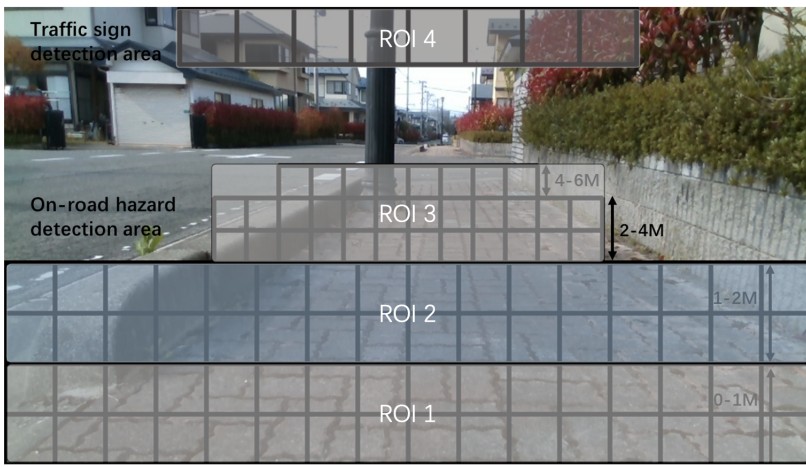

**Figure 4.** CoI configuration for hazard detection experiments. This figure demonstrates the allocation of 104 CoIs across four distinct RoIs on the image.

### 4.2. Indoor Obstacle Avoidance

The indoor obstacle avoidance experiment focused on detecting ground-level obstacles. This was different from the first experiment, where only specific areas were designated as CoIs. We distributed 30 irregular CoIs across the entire image to assess the network's performance. The detailed cell configuration is illustrated in Figure 5.

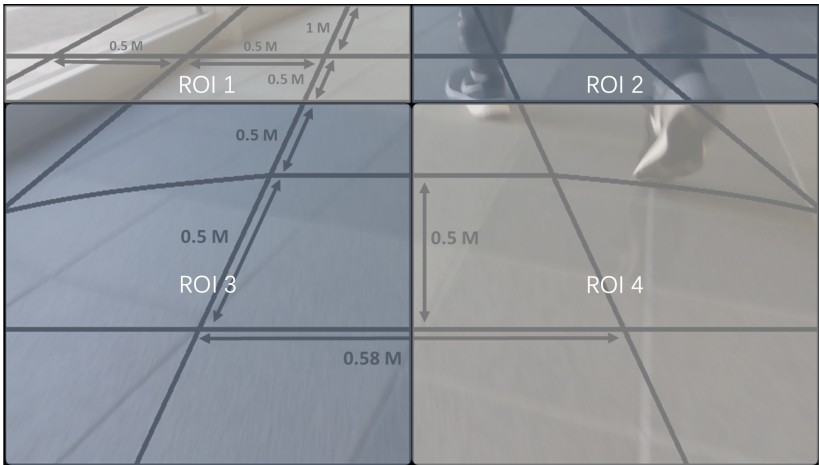

**Figure 5.** Configuration of CoIs for indoor obstacle avoidance. This illustration delineates 30 irregular CoIs allocated across four distinct RoIs on the floor, intended for the detection of ground-level obstacles.

For this experiment, we utilized an indoor image dataset comprising 6410 images featuring six common indoor objects: Sofa, Pillar, Door, Wall, People, and Other. The dataset was allocated across different phases of the experiment: 70% for training, 10% for validation, and the remaining 20% for testing. Due to the limited number of CoIs, we designated 16 CoIs above the images as RoI 1 and RoI 2, while 14 CoIs nearer to vehicles were classified

as RoI 3 and RoI 4. Correspondingly, a four-branch network was constructed and each branch was dedicated to one of these regions. Since the current object detection algorithms are not typically designed to handle irregular CoIs, our comparison is mainly between the proposed selective multi-branch network and the previous method [22]. This comparison aims to highlight the effectiveness of our selective multi-branch network in handling irregular CoI configurations. The results of this indoor experiment are detailed in Table 4. Additionally, we deployed our network on a Raspberry Pi 4B to compare the detection speeds, which are detailed in Table 5.

### 4.3. Performance Validation on Public Cityscapes Dataset

The Cityscapes dataset [40] is extensively used to evaluate the performance of semantic segmentation algorithms. Due to its detailed pixel-level annotations, we could effectively set up CoIs on the images and convert these labels according to the CoI positions. These images were categorized into 2975 for training, 500 for validation, and 1525 for testing. Because the Cityscapes dataset does not publicly provide annotations for test images, our models were trained and validated on the training set and then tested on the validation set.

As illustrated in Figure 6, we strategically positioned 256 CoIs around the vehicle area. This layout comprised 160 smaller CoIs centrally located to identify minute objects needing exact localization, complemented by 96 larger CoIs towards the image's lower section, enhancing the object detection near the vehicle. In this experiment, we utilized two branches to handle two RoIs at the image center, and four branches to detect objects for four RoIs in the lower area. This distribution was designed to ensure detailed and accurate detection in important areas. In these defined CoIs, we focused specifically on distinguishing people, vehicles, and the road. Consequently, we designed a selective multi-branch network with our CoI and RoI setup for model training.

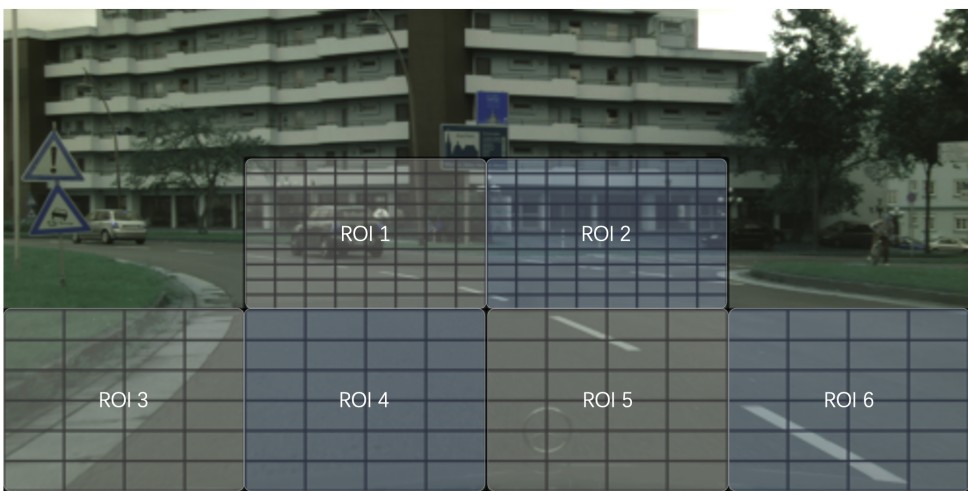

**Figure 6.** Deployment of CoIs and RoIs for vehicle and people detection. The configuration displays 256 CoIs distributed over six RoIs, with RoIs 1 and 2 positioned at the center for detailed object detection and RoIs 3 to 6 extending towards the lower section.

Table 6 provides a detailed comparison of our experimental results. We benchmark our selective multi-branch network against the YOLIC-M2 and YOLO models to demonstrate its effectiveness. Table 7 shows the speed comparison of the selective multi-branch model on the Cityscapes dataset.

**Table 4.** Performance comparison of different YOLIC models in terms of precision and recall for each obstacle category in the indoor environment experiment.

| Method | Input Size | Sofa | | | Wall | | | Pillar | | | People | | |
|---|---|---|---|---|---|---|---|---|---|---|---|---|---|
| | | Precision | Recall | $F_1$-Score | Precision | Recall | $F_1$-Score | Precision | Recall | $F_1$-Score | Precision | Recall | $F_1$-Score |
| YOLIC-M2 | 224 × 224 | 0.961 | 0.931 | 0.946 | 0.964 | 0.960 | 0.962 | 0.935 | 0.912 | 0.923 | 0.924 | 0.931 | 0.928 |
| Multi-branch | 224 × 224 | 0.966 | 0.893 | 0.928 | 0.967 | 0.940 | 0.953 | 0.932 | 0.863 | 0.897 | 0.939 | 0.909 | 0.924 |
| Selective multi-branch | 224 × 224 | 0.968 | 0.891 | 0.928 | 0.969 | 0.934 | 0.951 | 0.944 | 0.848 | 0.893 | 0.942 | 0.883 | 0.912 |

| Method | Input Size | Door | | | Other | | | All | | |
|---|---|---|---|---|---|---|---|---|---|---|
| | | Precision | Recall | $F_1$-Score | Precision | Recall | $F_1$-Score | Precision | Recall | $F_1$-Score |
| YOLIC-M2 | 224 × 224 | 0.941 | 0.942 | 0.941 | 0.942 | 0.875 | 0.907 | 0.945 | 0.925 | 0.935 |
| Multi-branch | 224 × 224 | 0.953 | 0.886 | 0.918 | 0.931 | 0.830 | 0.878 | 0.948 | 0.887 | 0.916 |
| Selective multi-branch | 224 × 224 | 0.953 | 0.884 | 0.918 | 0.936 | 0.821 | 0.875 | 0.952 | 0.877 | 0.913 |

**Table 5.** Evaluation of inference time (ms) on Raspberry Pi 4B with indoor obstacle dataset, including a selective multi-branch model with performance range indicated from minimum (no branch usage) to maximum load (full branch usage).

| Without Branch | One Branch | Two Branches | Three Branches | All Branches |
|---|---|---|---|---|
| 54.07 | 60.04 | 63.14 | 69.84 | 79.72 |

**Table 6.** Comparative performance metrics with different YOLO on Cityscapes dataset. Detection performance metrics for YOLO are sourced from [22].

| Method | Input Size | Vehicle | | | People | | | Other | | | All | | |
|---|---|---|---|---|---|---|---|---|---|---|---|---|---|
| | | Precision | Recall | $F_1$-Score | Precision | Recall | $F_1$-Score | Precision | Recall | $F_1$-Score | Precision | Recall | $F_1$-Score |
| YOLOv5-N | 640 × 640 | 0.873 | 0.856 | 0.864 | 0.778 | 0.780 | 0.779 | 0.919 | 0.687 | 0.786 | 0.857 | 0.774 | 0.813 |
| YOLOv5-S | 640 × 640 | 0.873 | 0.877 | 0.875 | 0.840 | 0.798 | 0.818 | 0.937 | 0.671 | 0.782 | 0.883 | 0.782 | 0.830 |
| YOLOv6-N | 640 × 640 | 0.825 | 0.594 | 0.691 | 0.703 | 0.635 | 0.667 | 0.462 | 0.471 | 0.466 | 0.663 | 0.567 | 0.611 |
| YOLOv8-N | 640 × 640 | 0.870 | 0.870 | 0.870 | 0.820 | 0.763 | 0.790 | 0.909 | 0.739 | 0.815 | 0.866 | 0.791 | 0.827 |
| YOLOv8-S | 640 × 640 | 0.884 | 0.860 | 0.872 | 0.820 | 0.839 | 0.829 | 0.942 | 0.670 | 0.783 | 0.882 | 0.790 | 0.833 |
| YOLIC-M2 | 224 × 224 | 0.880 | 0.839 | 0.859 | 0.745 | 0.616 | 0.674 | 0.917 | 0.929 | 0.923 | 0.847 | 0.795 | 0.820 |
| Multi-branch | 224 × 224 | 0.8853 | 0.8611 | 0.8730 | 0.7377 | 0.6607 | 0.6971 | 0.9238 | 0.9282 | 0.9260 | 0.8489 | 0.8167 | 0.8325 |
| Selective multi-branch | 224 × 224 | 0.8853 | 0.8606 | 0.8728 | 0.7377 | 0.6602 | 0.6968 | 0.9239 | 0.9276 | 0.9257 | 0.8490 | 0.8161 | 0.8318 |

**Table 7.** Evaluation of inference time (ms) on Raspberry Pi 4B with Cityscapes dataset, including selective multi-branch model, with performance range indicated from minimum (no branch usage) to maximum load (full branch usage).

| Without Branch | One Branch | Two Branches | Three Branches | Four Branches | Five Branches | All Branches |
|:---:|:---:|:---:|:---:|:---:|:---:|:---:|
| 56.32 | 58.91 | 62.57 | 67.17 | 72.87 | 76.27 | 80.92 |

To further illustrate the effectiveness of our proposed method, Figure 7 presents a visual comparison of the detection results obtained by our selective multi-branch network and the state-of-the-art YOLOv8-N algorithm. The sample images are selected from the outdoor hazard detection dataset and the Cityscapes dataset. As shown in the figure, our method accurately localizes and classifies objects within the predefined cells of interest, benefiting from the fixed CoI design. In contrast, YOLOv8-N, being a general object detection algorithm, may not always precisely localize objects within the specific areas of interest. This comparison highlights the advantage of our method in scenarios with fixed and known RoIs, resulting in more targeted and accurate detection.

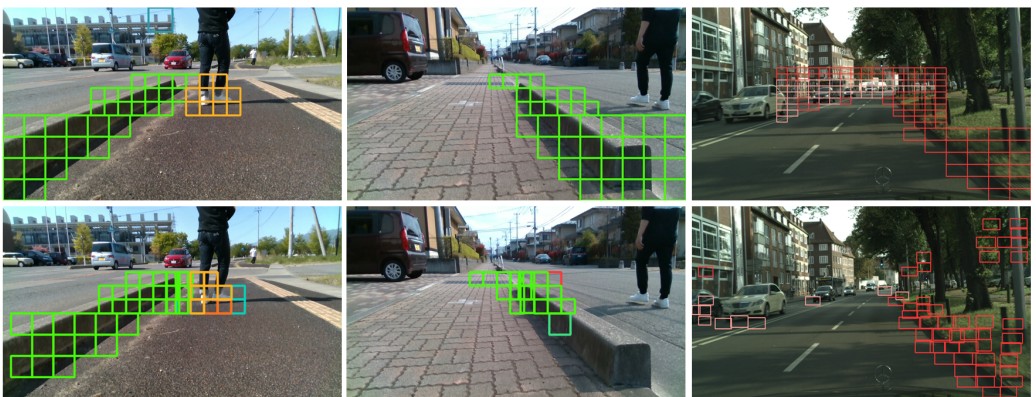

**Figure 7.** Some detection results obtained by the selective multi-branch network (first row) and the YOLOv8-N algorithm (second row) on sample images from the outdoor hazard detection dataset (first two columns) and the Cityscapes dataset (last column). Different colors of boxes represent different object classes. These results demonstrate that the proposed method can accurately localize and classify objects within the predefined cells of interest.

## 5. Discussions

In this section, we explore the comprehensive outcomes of our experiments with the proposed selective multi-branch network applied to three distinct detection tasks. A crucial aspect of this study involved evaluating the computational efficiency of our network. We compared the inference performance of several leading models on a Raspberry Pi 4B CPU. These models included YOLOv5-N [10], YOLOv6-N [11], YOLOv7-T [12], YOLOv8-N [13], NanoDet-m [41], MobileNet-SSD [7,42], and YOLIC-M2 [22]. The comparative analysis of the inference speeds can be found in Table 8. The expected inference time of the selective multi-branch model was calculated based on the following formula:

$$< T >= \frac{1}{Z} \sum (p_i \times T_i) \qquad (9)$$

where $Z = \sum(p_i)$ captures the sum of the number of images in the evaluation dataset, $p_i$ represents the count of images that require the activation of $i$ branches for inference, and $T_i$ denotes the time needed for the model to perform inference using $i$ branches.

Our first experiment focused on outdoor hazard detection, particularly designed for low-cost electric scooters. According to the data presented in Table 2, our selective multi-

branch network outperformed all standard object detection algorithms, achieving an $F_1$ score of 0.8524, and this performance is closely comparable to that of the YOLIC network and multi-branch network. In scenarios without detectable risks (where sub-branches were not activated), the network required only 0.35G FLOPs and achieved an impressive inference time of just 59 ms on the Raspberry Pi 4B CPU. This was the fastest among all models tested, highlighting the network's suitability for real-time applications in outdoor environments. Furthermore, even when all sub-branches were activated, the selective multi-branch network maintained a competitive inference time of 79 ms. Figure 8a presents the relationship between the number of branches required for network inference and the number of images in the outdoor hazard detection dataset. It is important to note that scenarios wherein an image necessitates the use of all branches are exceedingly rare. Our selective multi-branch demonstrates its efficiency, with an average inference time of 64.19 ms, significantly faster than the YOLIC network's 74.98 ms and the multi-branch model's 74.82 ms.

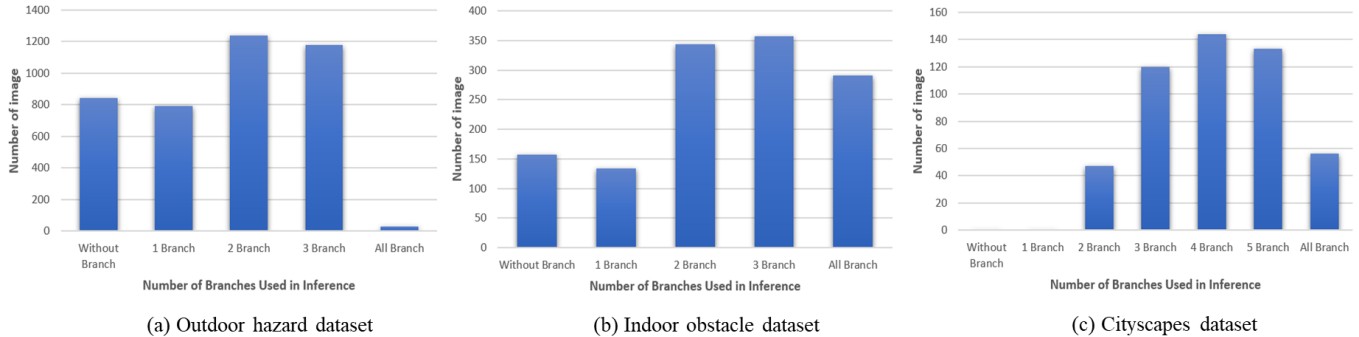

(a) Outdoor hazard dataset       (b) Indoor obstacle dataset       (c) Cityscapes dataset

**Figure 8.** These figures present the distribution of the required number of branches for network inference across three different datasets. The horizontal axis represents the number of branches utilized in network inference, while the vertical axis quantifies the count of images necessitating a specific branch quantity.

**Table 8.** Inference time evaluation on Raspberry Pi 4B across different models using the ncnn framework, notably optimized for mobile platforms. Results for our selective multi-branch network are highlighted in bold. All models were tested using FP16 precision.

| Model | Input Size | #Params | FLOPs (G) | Inference Time (ms) |
|---|---|---|---|---|
| YOLOv5-N | 640 × 640 | 1.9 M | 4.5 | 529.68 |
| YOLOv6-N | 640 × 640 | 4.7 M | 11.4 | 500.30 |
| YOLOv7-T | 416 × 416 | 6.2 M | 5.8 | 230.53 |
| YOLOV8-N | 640 × 640 | 3.2 M | 8.7 | 499.38 |
| NanoDet-m | 320 × 320 | 0.95 M | 0.72 | 81.69 |
| MobileNet-SSD | 300 × 300 | 6.8 M | 2.50 | 137.40 |
| YOLIC-M2 (Outdoor) | 224 × 224 | 3.8 M | 0.62 | 74.98 |
| YOLIC-M2 (Indoor) | 224 × 224 | 2.5 M | 0.62 | 73.29 |
| YOLIC-M2 (Cityscapes) | 224 × 224 | 3.5 M | 0.62 | 74.68 |
| Multi-branch (Outdoor) | 224 × 224 | 9.8 M | 0.58 | 74.82 |
| Multi-branch (Indoor) | 224 × 224 | 8.48 M | 0.70 | 79.72 |
| Multi-branch (Cityscapes) | 224 × 224 | 13.49 M | 0.56 | 80.92 |
| **Selective multi-branch (Outdoor)** | **224 × 224** | **9.8 M** | **0.35–0.58** | **64.19** |
| **Selective multi-branch (Indoor)** | **224 × 224** | **8.48 M** | **0.35–0.70** | **67.33** |
| **Selective multi-branch (Cityscapes)** | **224 × 224** | **13.49 M** | **0.35–0.56** | **72.33** |

In the indoor obstacle avoidance experiment, we tackled the challenge of detecting ground-level obstacles using a non-rectangular CoI configuration. This CoI configuration precluded direct comparisons with traditional object detection algorithms. However,

as evidenced in Table 4, the selective multi-branch network achieved results comparable to the YOLIC network. The speed evaluation in Table 5 further reinforces the network's capability. Even under the non-standard CoI setup, our network surpassed the traditional object detection algorithms in terms of inference speed. According to Figure 8b, the majority of images within the indoor obstacle dataset required the activation of only 2–3 branches for successful inference, showcasing the network's efficiency. Remarkably, the selective multi-branch network not only surpassed the YOLIC network with the fastest recorded inference time of 67.33 ms but also exceeded the multi-branch network's time of 79.72 ms.

Lastly, our analysis on the Cityscapes dataset presents encouraging results. As presented in Table 6, the selective multi-branch network achieved an $F_1$ score of 0.8318, surpassing the single MobileNetV2-based YOLIC network (0.8202) and closely matching YOLOv8-S (0.8333). This outcome indicates the network's effectiveness in urban scene analysis tasks. Figure 8c reveals that in the complex Cityscapes dataset, a mere 11% of the scenarios necessitated the activation of all branches, with the majority of cases requiring just 3–5 branches for inference. The inference times detailed in Table 8 further illustrate the network's agility on the public dataset.

From the above experiments, we can see that our proposed selective multi-branch network not only achieves competitive performance but also significantly reduces the computational overhead and increases the inference speed. These improvements make our method potentially suitable for a wide range of real-world applications. For example, the proposed selective multi-branch network can be used for real-time obstacle detection and avoidance in low-cost vehicles, such as delivery robots, electric scooters, and drones. Our selective multi-branch network can also be applied to traffic monitoring, where different branches can be assigned to monitor specific road sections. Furthermore, in parking lot surveillance systems, our method can be utilized to monitor vehicle occupancy, with individual branches allocated to monitor specific parking sections. These diverse applications demonstrate the practicality and versatility of our proposed selective multi-branch network.

In summary, our selective multi-branch network exhibits varying detection speeds depending on the input scenario, achieving the fastest inference times among all networks in optimal conditions. However, due to limitations in our current code implementation, the branches of our network operate sequentially. If these activated branches could be executed in parallel, it would further enhance the network's operational speed. The main reason for using this network on mobile devices is because it saves computing power and battery life, especially when the device is in standby mode. This flexibility makes the device work better and last longer in real-life use.

## 6. Conclusions

In this study, we have introduced a selective multi-branch network for real-time object localization and classification on edge devices. The proposed network incorporates a selective attention unit (SAU) and a selective feature map fusion mechanism to reduce the computational overhead and improve the inference speed while maintaining competitive performance. The experimental results show that the newly proposed network can be 2.35 ms, 5.96 ms, and 10.79 ms faster (on average) than the original YOLIC-M2 model for the Cityscapes, indoor, and outdoor datasets, respectively. In practical driving scenarios, we usually focus on regions in front of the vehicle, and we check the side mirrors with a low frequency. Thus, if we design the SAU to emulate a human driver, the average number of selected branches for decision making will be much smaller and the reduction in the inference time will be more significant. Moreover, our network achieved faster inference speeds compared to the state-of-the-art YOLOv8-N model on the Raspberry Pi 4B.

However, the proposed method has some limitations. The main limitation is that once the regions of interest (RoIs) and cells of interest (CoIs) are defined and fixed, it becomes difficult to flexibly adjust them when the detection needs and locations change in later stages. To address this limitation, we plan to propose a mechanism that can dynamically and adaptively re-configure the RoIs and CoIs based on environmental changes. In our

future work, we also plan to investigate and optimize the activation thresholds for the SAU to select the branches based on the input data and the desired balance between accuracy and efficiency.

**Author Contributions:** Conceptualization, K.S.; methodology, K.S.; validation, K.S.; writing—original draft preparation, K.S.; writing—review and editing, K.S., Y.T., Q.Z. and Y.L.; supervision, Y.T., Q.Z. and Y.L. All authors have read and agreed to the published version of the manuscript.

**Funding:** This research received no external funding.

**Institutional Review Board Statement:** Not applicable.

**Informed Consent Statement:** Not applicable.

**Data Availability Statement:** The datasets used in this study are available at the following Kaggle repositories. Outdoor hazard detection dataset: https://www.kaggle.com/datasets/sukai3316/outdoor-hazard-detection-dataset (accessed on 10 April 2024). Indoor obstacle avoidance dataset: https://www.kaggle.com/datasets/sukai3316/indoor-obstacle-avoidance-dataset (accessed on 10 April 2024).

**Acknowledgments:** We thank the cooperating research company, which provided us with the equipment necessary to conduct the experiments.

**Conflicts of Interest:** The authors declare no conflicts of interest.

**Abbreviations**

The following abbreviations are used in this manuscript:

| | |
|---|---|
| YOLIC | You Only Look at Interested Cells |
| CoIs | Cells of Interest |
| RoIs | Regions of Interest |
| MAC | Multiply-Accumulate Operation |
| SAU | Selective Attention Unit |
| FPN | Feature Pyramid Network |

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
