# Peer review of "A Selective Multi-Branch Network for Edge-Oriented Object Localization and Classification"

_electronics, doi:10.3390/electronics13081472_

Round 1

Reviewer 1 Report

Comments and Suggestions for Authors

(1) For the improvement of the algorithm proposed in this article, is it calculated at the edge or in the cloud? Does it exist in the form of an API?

(2) What other reviews are there besides Raspberry Pi 4B? Does this technology support real-time image recognition acquired in time series?

(3) As we all know, the analysis of any data relies on learning and modeling. Have the authors considered any practical applications of the results of this research in different scenarios?

(4) The conclusion chapter should be rewritten. The authors should clearly state the contributions and limitations of this paper as well as directions for future work.

(5) The author should add at least 10-20 of the latest references published after 2021.

That's all. Thanks. 

Author Response

Dear Reviewer,

Thank you very much for dedicating your time to review our manuscript. We truly appreciate all your generous comments and suggestions. In response to your valuable and insightful comments, we have made improvements to the current version. We have carefully considered each comment and have done our best to address every one of them. Please find attached our detailed responses to your comments.

Thank you once again for your invaluable contribution to our work.

Yours sincerely,

Kai Su

Reviewer 2 Report

Comments and Suggestions for Authors

1.Authors should provide datasets that they have used in outdoor and indoor experiments for readers to verify the algorithm.

2. The lightweight models YOLOv5 and YOLOv3-tiny mentioned in the literature review are not the latest technologies. Please review the research progress of the latest lightweight models.

4.Could you provide some insight into why you chose 0.5 as the threshold for activating sub-branches? Were there specific considerations or experiments that led to this choice?

5.Please add a description of how each feature map is cropped based on the RoI.

6. It is suggested to provide an example of target recognition in the experimental part, and compare it with other algorithms to show the accuracy of the algorithm designed by the author

7.Please mention the limitations of your current work and future directions in your conclusion or in a dedicated paragraph.

Author Response

(The authors gave the same response as above.)

Round 2

Reviewer 1 Report

Comments and Suggestions for Authors

(1) In the experiments in Section 4, the author used more than 70% of the data for training for each case study, and the remaining data was used for verification and testing. What is the basis for this data proportional division? Do this study's results rely on bias in the data itself?

(2) Are the methods of the YOLO series all free and open source? What are the difficulties developers face in exploiting them?

(3) The reviewer still believes that the author should propose a framework diagram that includes the overall concept of PyTorch and the role of hardware (such as an NVIDIA RTX 4090 GPU and Raspberry Pi 4B) in the experiment.

That is all. Thanks.

Author Response

Dear Reviewer,

Thank you very much for dedicating your time to review our manuscript. We truly appreciate all your generous comments and suggestions. In response to your valuable and insightful comments, we have further refined our manuscript. We have meticulously addressed each of your comments, ensuring that no point was overlooked. Please find attached our detailed responses to your comments.

Thank you once again for your invaluable contribution to our work.

Yours sincerely,

Kai Su
